🔓 | **Open Peer Review** | Applied and Industrial Microbiology | Research Article

# Oxygen evolution from extremophilic cyanobacteria confined in hard biocoatings

**Simone Krings,[1] Yuxiu Chen,[2] Joseph L. Keddie,[2] Suzanne Hingley-Wilson[1]**

**ABSTRACT** Biocoatings, in which viable bacteria are immobilized within a waterborne polymer coating for a wide range of potential applications, have garnered greater interest in recent years. In bioreactors, biocoatings can be ready-to-use alternatives for carbon capture or biofuel production that could be reused multiple times. Here, we have immobilized cyanobacteria in mechanically hard biocoatings, which were deposited from polymer colloids in water (i.e., latex). The biocoatings are formed upon heating to 37°C and fully dried before rehydrating. The viability and oxygen evolution of three cyanobacterial species within the biocoatings were compared. *Synechococcus* sp. PCC 7002 was non-viable inside the biocoatings immediately after drying, whereas *Synechocystis* sp. PCC 6803 survived the coating formation, as shown by an adenosine triphosphate (ATP) assay. *Synechocystis* sp. PCC 6803 consumed oxygen (by cell respiration) for up to 5 days, but was unable to perform photosynthesis, as indicated by a lack of oxygen evolution. However, *Chroococcidiopsis cubana* PCC 7433, a strain of desiccation-resistant extremophilic cyanobacteria, survived and performed photosynthesis and carbon capture within the biocoating, with specific rates of oxygen evolution up to 0.4 g of oxygen/g of biomass per day. Continuous measurements of dissolved oxygen were carried out over a month and showed no sign of decreasing activity. Extremophilic cyanobacteria are viable in a variety of environments, making them ideal candidates for use in biocoatings and other biotechnology.

**IMPORTANCE** As water has become a precious resource, there is a growing need for less water-intensive use of microorganisms, while avoiding desiccation stress. Mechanically robust, ready-to-use biocoatings or "living paints" (a type of artificial biofilm consisting of a synthetic matrix containing functional bacteria) represent a novel way to address these issues. Here, we describe the revolutionary, first-ever use of an extremophilic cyanobacterium (*Chroococcidiopsis cubana* PCC 7433) in biocoatings, which were able to produce high levels of oxygen and carbon capture for at least 1 month despite complete desiccation and subsequent rehydration. Beyond culturing viable bacteria with reduced water resources, this pioneering use of extremophiles in biocoatings could be further developed for a variety of applications, including carbon capture, wastewater treatment and biofuel production.

**KEYWORDS** biocoatings, extremophiles, cyanobacteria, *Chroococcidiopsis cubana* PCC 7433, oxygen evolution, carbon dioxide fixation

With the increase of greenhouse gases, particularly carbon dioxide ($CO_2$), in the atmosphere and the accompanying climate crisis, interest in the use of cyanobacteria for $CO_2$ sequestration has grown (1–4). Cyanobacteria and microalgae fix $CO_2$ to transform it via photosynthesis into organic compounds with high efficiency and can survive in adverse environments (e.g., saline-alkaline water) (4, 5). Additionally, these microorganisms grow quickly and can be readily genetically modified in most cases (4).

Address correspondence to Suzanne Hingley-Wilson, s.hingley-wilson@surrey.ac.uk.

The authors declare no conflict of interest.

See the funding table on p. 17.

For many applications, high densities of cyanobacterial biomass are needed. Biorefineries seek to employ single dense cultures for the production of biofuels (biodiesel, biogas, biohydrogen, bioethanol, and biobutanol) and other value-added products (e.g., carotenoids) simultaneously (4, 6, 7). However, phenotypic instability, wherein non-producing phenotypes emerge and outcompete the producing cells in industrial scale bioreactors, has been reported as a major issue (8).

Many uses, however, do not require the high-density cultivation of bacteria, e.g., in the production of ethanol or hydrogen (6, 9, 10). As a result, alongside using bioreactors, the immobilization of microorganisms in matrices has garnered interest over the past years. As a physical barrier between the cells and the medium, an immobilizing material eases the recovery of products, reduces the volume of liquid medium needed, and maintains high cell concentrations (9). For example, the secretion of glucose and sucrose from cyanobacteria or algae fermented under anaerobic dark fermentation to produce bioethanol could be collected from the medium directly in such a model, thus avoiding the need for biomass extraction (6). Previous studies already proved hydrogen production by immobilized *Rhodopseudomonas palustris* (10, 11). In addition, phenotypic instability would be reduced, as cells are under growth arrest or growing slowly (8). One option is to confine the bacteria within coatings, called "biocoatings" (biocatalytic coatings), a term coined by Flickinger et al. to refer to bacteria immobilized within a polymer coating made from waterborne particles (9, 12, 13).

Synthetic waterborne polymer colloids, called latex, are produced via the emulsion polymerization of monomers, such as acrylates (e.g., methyl methacrylate or butyl acrylate) or styrene, and typically stabilized by charge or steric repulsion. To create biocoatings, a mixture of latex, bacteria, and other components (e.g., sugars or nano-fillers) in water are deposited on any substrate (Fig. 1, Step 1). The water evaporates, and the latex particles pack closely together (Step 2) (14). At a temperature above the latex polymer's glass transition temperature ($T_g$), the particles deform to fill space (Step 3) (14, 15). Molecules in the individual latex particles diffuse across boundaries to achieve coalescence into a cohesive coating (Step 4) (14). The particle deformation can be arrested to avoid full coalescence, resulting in a porous coating (16).

Since the first development of biocoatings, there has been a need to create and retain permeability in order to allow hydration and nutrient transport to maintain cell viability. Early examples of formulations contained carbohydrates creating pores when dissolved in the rehydration liquid (10). However, these porous structures were not structurally stable because of continuing coalescence of their constituent colloidal particles (9, 12, 17, 18). Recent research from our group described the addition of halloysite nanoclay to create a network of pores in biocoatings to increase permeability and, hence, the viability of *Escherichia coli* (16) and nitrifying bacteria (19).

The immobilization of different strains of cyanobacteria within biocoatings was described by Bernal et al. (18). They successfully immobilized four strains on chromatography paper substrates without drying the cell formulation and then immediately rehydrating by capillary action without suspension in bulk liquid. Because of the low $T_g$ of the latex, film formation with full particle coalescence resulted in low permeability of the biocoating and, hence, low cell viability (12, 17, 18). Moreover, in latex with a $T_g$ below the ambient temperature, the coatings lack abrasion resistance and durability (9, 20). However, Lyngberg et al. created porous biocoatings with hyperthermophile *Thermotoga maritima* using high-$T_g$ particles as hard fillers and the low-$T_g$ particles to create a continuous binder phase, which allowed their utilization of glucose and maltose, therefore proving viability (17).

The aim of our research was to immobilize metabolically active cyanobacteria in a mechanically hard coating to fix carbon and to evolve oxygen. The first objective, from the materials' point of view, was to create porous yet mechanically robust and hard biocoatings. Processing of these coatings requires heating to temperatures above the copolymer's $T_g$. When cooled to a temperature below the $T_g$, the biocoating is expected to remain porous because particle coalescence will not continue (16).

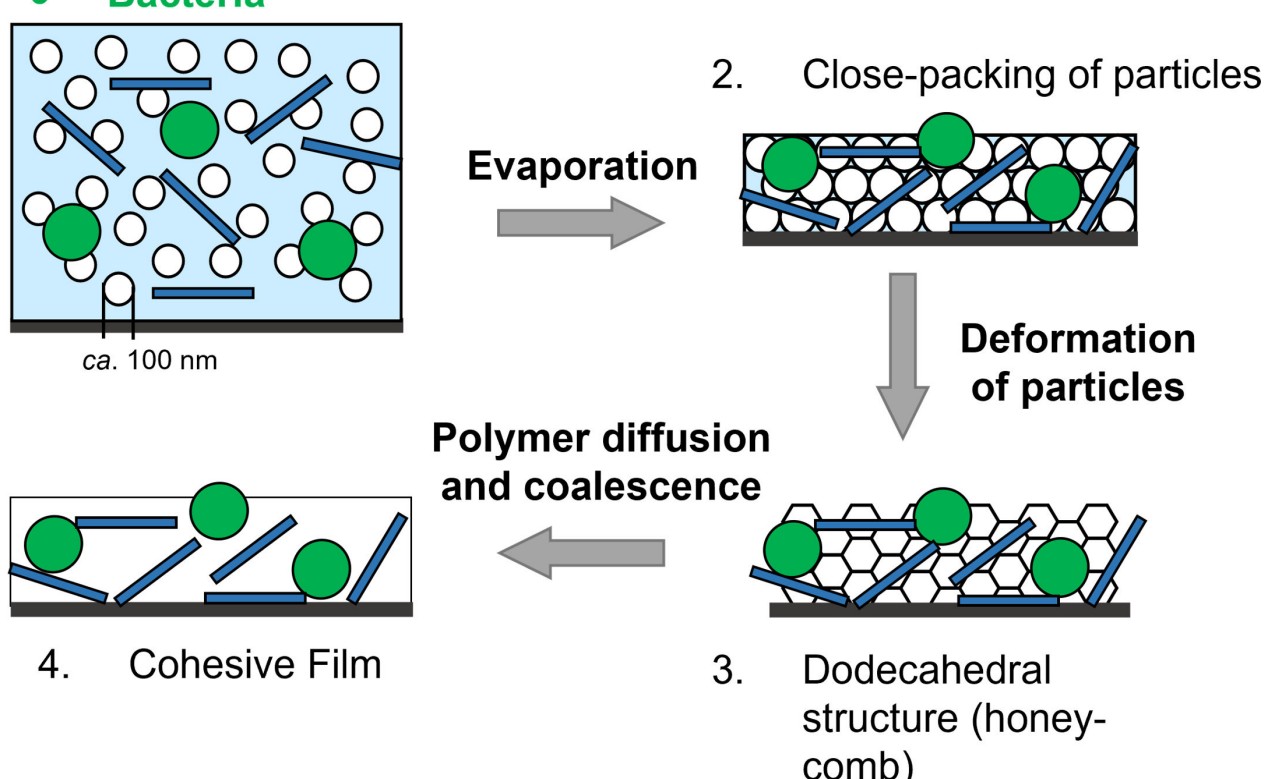

**FIG 1** The film formation process of the biocoating formulation, consisting of four steps. (1) The aqueous biocoating mixture, containing latex and bacteria, is deposited on a substrate. (2) The evaporation of the water takes place and the particles become closely packed (ideally in a face-centered cubic structure). (3) The particles deform to fill space and adopt a rhombic dodecahedral structure, which results in optical clarity of the coating. (4) The polymer molecules diffuse across the particle boundaries, and the particles coalesce when at a temperature above the glass transition temperature ($T_g$). A cohesive film results. (Diagram not drawn to scale.)

The second objective was to maintain the viability of the cyanobacteria within the biocoatings, as the formation process requires the bacteria to be desiccation-tolerant. The marine strain *Synechococcus* sp. PCC 7002 was chosen for its tolerance to NaCl and high levels of light and its fast doubling time. Furthermore, it is naturally transformable, which could be of use for future applications (21, 22). Also chosen was *Synechocystis* sp. PCC 6803, which is a freshwater-derived model organism that can also resist high salt environments. It can be genetically engineered to withstand biofuels and ethanol (23, 24) and was also able to produce hydrogen when immobilized in alginate beads (25). The third choice is *Chroococcidiopsis cubana* PCC 7433 (SAG39.79), which was first isolated from a dried pool in Cuba and is part of a genus that has extremophilic features, e.g., desiccation- and radiation-resistance (26–29). As an example, Fagliarone et al. demonstrated that members of this genus were able to survive air-dried storage for 4 years (26). This genus is known to inhabit and thrive on the surface of rocks and is a potential candidate for Mars colonization (28, 30, 31). A better understanding of their desiccation resistance could lead to a range of applications for biotechnology (32, 33).

Cyanobacterial biocoatings could be utilized in the production of biofuels, such as ethanol or hydrogen, which do not depend on the extraction of the bacteria as many strategies do (6, 9, 11). In addition to carbon capture, they could also have applications

in bioremediation, e.g., in wastewater treatment or in contaminated soils (34). As water is a precious resource, transporting and storing the biocoatings dry and rehydrating them upon use would be a major advantage over conventional culture transport and set-up (9). They could also prove beneficial in environments where growth medium supply is low, for example, in space stations (31).

## RESULTS

### Assessing the coating hardness

The hardness of the latex polymer coating via the pendulum damping time was evaluated to establish whether it would be suitable for applications subjecting it to mechanical abrasion. The König pendulum hardness of the latex coating ($T_g$ = 34°C) was measured to be 115.6 ± 5.5 s at a temperature of 20°C. A comparator "softer" acrylic latex ($T_g$ = 20°C) had a lower hardness of 38.7 ± 1.2 s ($n$ = 12, Mean ± SD), which allowed it to be easily indented with a fingernail. As a second comparison, the hardness of glass was measured to be higher at 257.6 s (Fig. 2). The latex coating with a $T_g$ of 34°C is less hard than glass but still offers mechanical durability. The Young's modulus of the copolymer with $T_g$ = 34°C was previously reported (16) to be 562 ± 24 MPa. With the addition of halloysite (20 vol.%), the modulus was reduced and the coating became more flexible (16).

### Evaluating the toxicity of the latex suspensions

To evaluate the toxicity of the latex suspensions to the bacteria, the Miles and Misra plate count method was used (35). The latex was toxic to *Synechococcus* 7002 (****$p$< 0.0001) and diluting the latex with (LH) or without (LW) halloysite resulted in significantly higher survival (Fig. 3A). This was in contrast to the bacterial counts of *Synechocystis* 6803 on D0 and D7, which showed no statistically significant difference within each type of sample and were, therefore, non-toxic (Fig. 3B). In the case of *Chroococcidiopsis* 7433, there was a less than 1-log drop in viability in the latex samples (*$p$= 0.0161). However, this was not observed in the diluted samples (LH and LW) in which the bacterial numbers remained stable over the tested period of 7 days (Fig. 3C).

Although the latex was toxic in varying degrees to two of the cyanobacterial species (*Synechococcus* 7002 and *Chroococcidiopsis* 7433), dilution with water only (LW) or halloysite suspensions (LH) dramatically reduced the toxicity. The addition of halloysite nanotubes did not show any detrimental effects to the survival of the bacteria, which proves their biocompatibility in suspension at this working concentration based on our prior work (16).

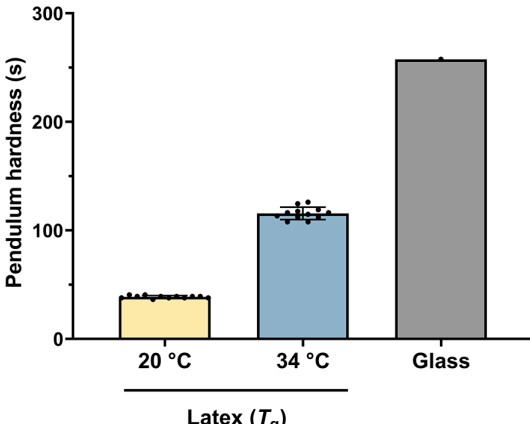

**FIG 2** König Pendulum hardness (s) of a low-$T_g$ latex ($T_g$ = 20°C), a high-$T_g$ latex ($T_g$ = 34°C), and glass (as a comparator) at a temperature of 20°C. The low-$T_g$ latex has a lower pendulum hardness, which highlights its relative softness, while the high-$T_g$ latex—used in the following experiments—was harder ($n$ = 12, Mean ± SD).

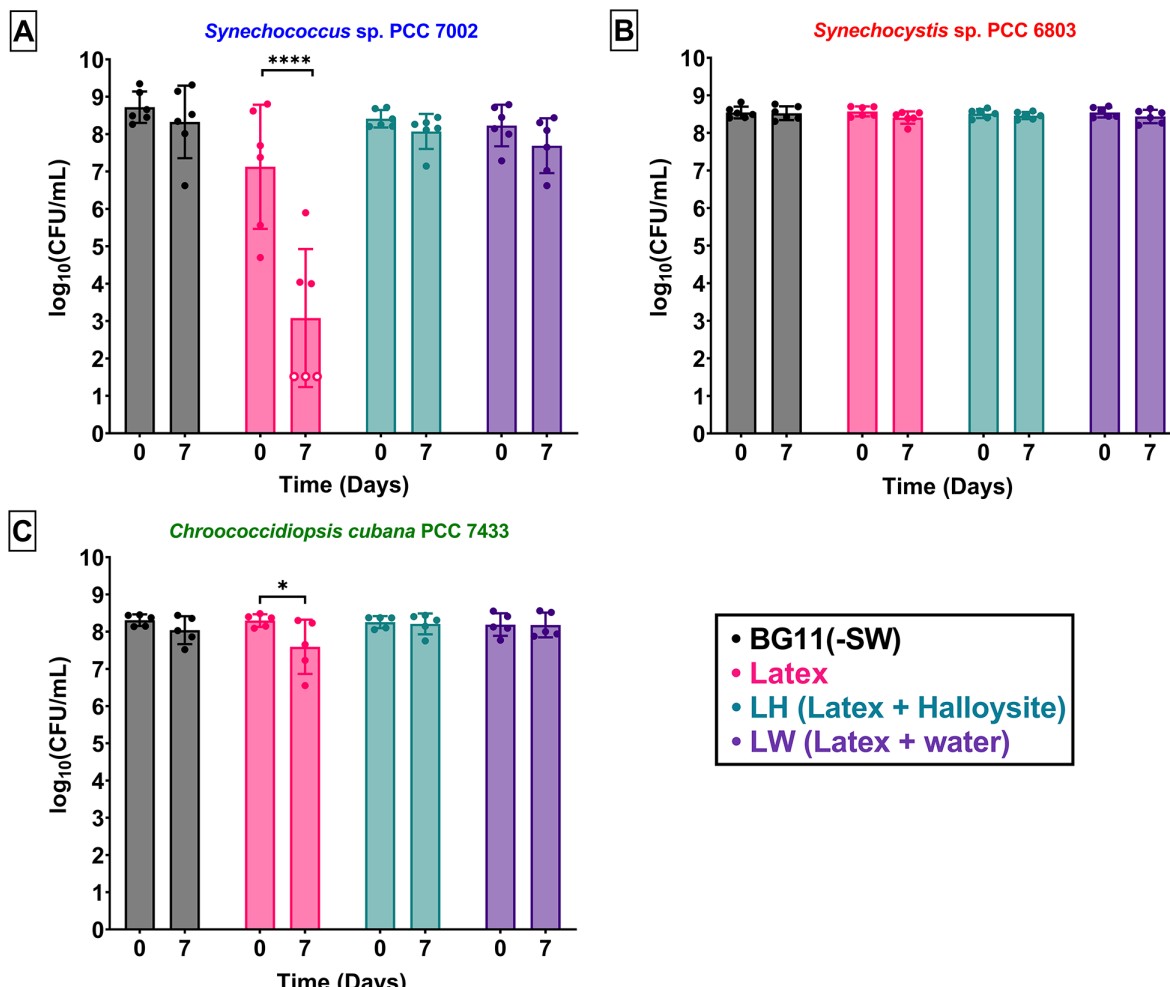

**FIG 3** The biocoating formulation of interest (LH) was not toxic to the cyanobacteria prior to film formation. Toxicity of the latex suspensions to the three strains of cyanobacteria tested, (A) *Synechococcus* sp. PCC 7002, (B) *Synechocystis* sp. PCC 6803, and (C) *Chroococcidiopsis cubana* PCC 7433, on Day 0 and after 7 days at room temperature (RT) in the dark (Day 7). Statistically significant differences between Day 0 and Day 7 could be observed in latex suspensions containing *Synechococcus* 7002 (****$p < 0.0001$) [three values were below the limit of quantification (open circle)] and *Chroococcidiopsis* 7433 (*$p = 0.0161$). No decrease in viability could be observed in the other samples or in any samples containing *Synechocystis* 6803 (ns). [Mean ± standard error of the mean (SEM), two-way ANOVA followed by Šídák's multiple comparisons test, (A) and (B) $n = 6$ and (C) $n = 5$ biological replicates for all samples.]

## Visualization of the biocoating structure and chemical identification of the components

### Scanning electron microscopy and energy-dispersive X-ray spectroscopy

Scanning electron microscopy (SEM) was used to confirm the successful immobilization and even distribution of cyanobacteria in the biocoatings. First, the microscopic morphology of the individual biocoating components was determined to help identify them when mixed. The latex particles have a round shape, whereas the halloysite nanoclay is tubular and needle-shaped [Fig. 4A (a) and (b), respectively]. *Synechococcus* 7002 is rod-shaped, while *Synechocystis* 6803 and the larger *Chroococcidiopsis* 7433 are coccoid, as expected. Cell division could also be observed [Fig. 4A (c–e)]. Within the biocoatings, the respective cyanobacteria (rod-shaped *Synechococcus* 7002, coccoid *Synechocystis* 6803, and larger *Chroococcidiopsis* 7433) and halloysite nanotubes could be identified [Fig. 4A (f–h)].

To further prove the identification of individual components in this complex biocoating structure, energy-dispersive X-ray spectroscopy (EDX) was carried out to

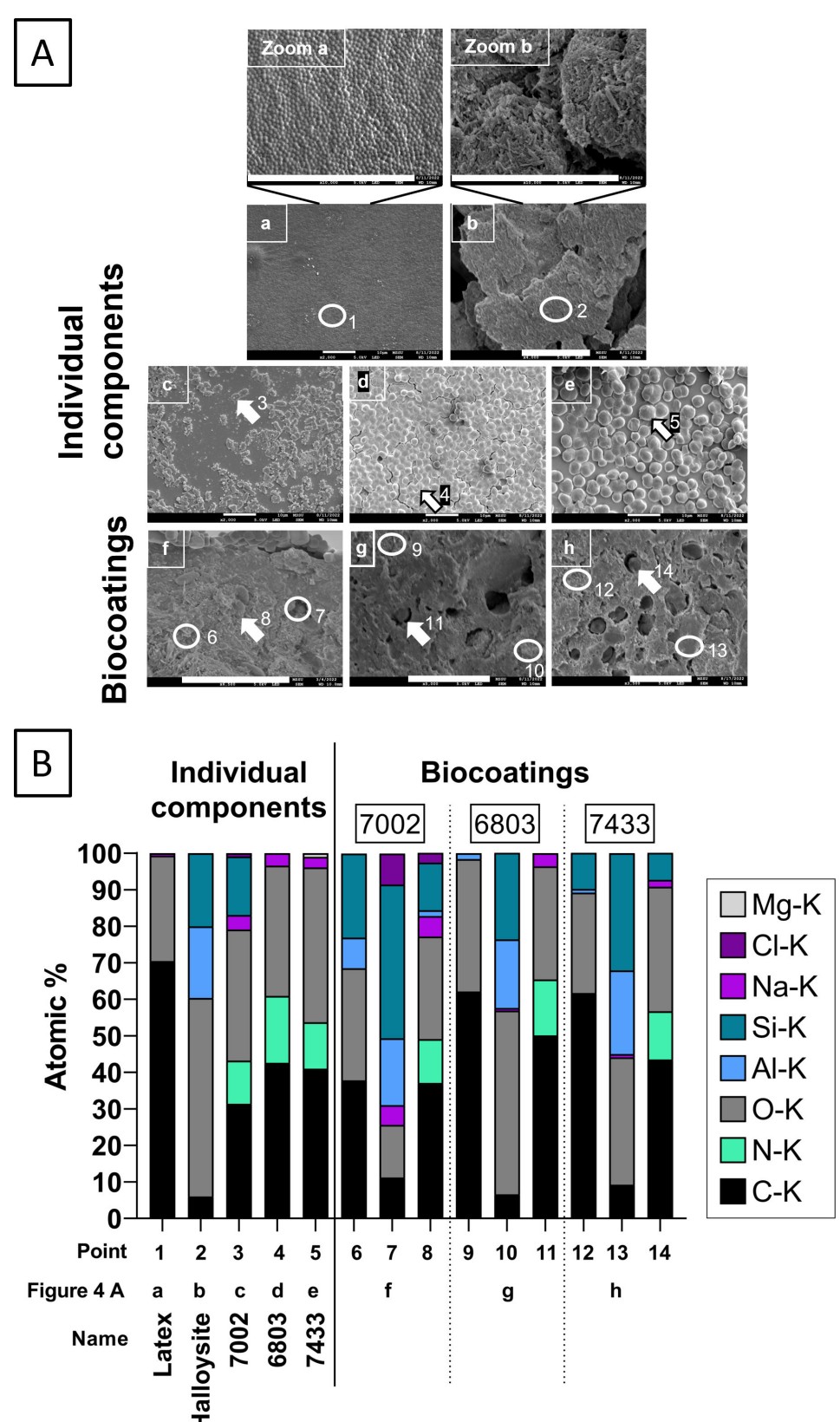

**FIG 4** Identification of the biocoating components by microscopy and elemental analysis. SEM/energy-dispersive X-ray spectroscopy (EDX) of individual components and film-formed, cross-sectioned biocoatings. (A) SEM images of separate biocoating components (a–e, as controls): (a and Zoom a) latex; (b and Zoom b) halloysite; (c) *Synechococcus* sp. PCC 7002; (Continued on next page)

**FIG 4** (Continued)

(d) *Synechocystis* sp. PCC 6803; (e) *Chroococcidiopsis cubana* PCC 7433; and after film formation and rehydration of biocoatings: (f) *Synechococcus* sp. PCC 7002; (g) *Synechocystis* sp. PCC 6803; (h) *Chroococcidiopsis cubana* PCC 7433. Scale bars: 10 µm. Points marked by arrows indicate cyanobacteria and circles indicate latex and halloysite, which were analyzed by EDX. (B) Representative points were analyzed by EDX. Atomic % are represented. (a) Latex is composed of C and O; (b) halloysite is composed by O, Al, and Si, as well as C from the poly(ethylene glycol) (PEG) addition; (c–e) cyanobacteria consist of C, O, N, and Na. N could only be observed in the cyanobacterial samples. The observed Si in (c) is attributed to the glass substrate. (f–h) Cross-sectioned biocoatings: (f) was cryo-fractured, while (g) and (h) were cryo-sectioned. Point (area of interest, AOI) 6 was a mixture of latex and halloysite, whereas Points 9 and 12 are more similar in composition to the latex. Points 7, 10, and 13 are mostly halloysite. Points 8, 11, and 14 are bacteria-shaped in the SEM images and, indeed, have a similar composition to the bacteria on their own.

determine the differential elemental compositions of each component (Fig. 4B). A total of 14 area of interest (AOI) were analyzed. AOI 1–5 are from the individual components, which act as standards/controls. AOI 6–14 are small areas on the surface of the biocoatings cross section, where certain components had been identified based on their morphology.

The acrylic copolymer is expressed at 75 at.% C and 25 at.% O, as H is undetectable by EDX. The experimental detection of 70 at.% C and 29 at.% O for this phase (AOI 1) is attributed to small amounts of surfactants and initiators in the sample. The halloysite mineral sample additionally contains C and O because of the poly(ethylene glycol) (PEG) stabilization. The calculated composition of the dried halloysite suspension is 4.76 at.% C, 66.67 at.% O, 14.29 at.% Al, and 14.29 at.% Si, which is similar to the experimentally found composition of 5.98 at.% C, 54.38 at.% O, 19.61 at.% Al, and 20.03 at.% Si (AOI 2). Cyanobacteria consist of C, O, N, and trace amounts of Na (AOI 3–5). As N was absent from both latex and halloysite and only present in bacterial samples, it was used as a marker for their identification in the biocoatings. The Si found in the analysis of *Synechococcus* 7002 can be attributed to the glass substrate.

Following the determination of the elemental composition of the individual components, it was possible to confirm the composition of the complex biocoatings. AOI 7, 10, and 13 contained high levels of Al and Si, which suggests these areas were rich in halloysite. AOI 6 displayed lower levels of Al and Si and AOI 9 and 12 a near-complete absence of them and, therefore, only the presence of latex particles. AOI 8, 11, and 14 contained N, the signature of cyanobacteria, therefore confirming the successful immobilization of cyanobacteria. EDX can be a powerful tool to identify the different components of biocoatings without the need for added markers, offering the means to study more complex structures and compositions in the future (see EDX spectra in Supplemental Material).

## Confocal laser scanning microscopy

Thanks to their naturally occurring pigments, cyanobacteria could be visualized within the biocoatings without the use of additional dyes or fluorescent tags (Fig. 5). As previously described, the presence of chlorophyll *a* and phycobilisomes could be detected thanks to their red fluorescence (e.g., in viable cells), whereas the degradation of chlorophyll *a* revealed the underlying secondary pigments, fluorescing green (e.g.,in dying cells) (36–38). Although not all lethal stress leads to the degradation of the light-harvesting pigments, confocal laser scanning microscopy (CLSM) was used to assess any visible differences in biocoatings using the different strains. In addition, their characteristic morphology could be observed. As expected, *Synechocystis* 6803 (~2 µm) and the larger *Chroococcidiopsis* 7433 (~5 µm) are of coccoid shape, whereas *Synechococcus* 7002 (~2 µm) are rod-shaped. The visualization of the bacteria within the biocoatings was slightly more difficult due to heterogeneities in the coatings and the associated scattering of light. However, for all three strains (Fig. 5A, B and C), the majority of cells appeared red, meaning they still contained chlorophyll *a* and phycobilisomes and appeared viable within the biocoatings. The fluorescence of the pigments can be

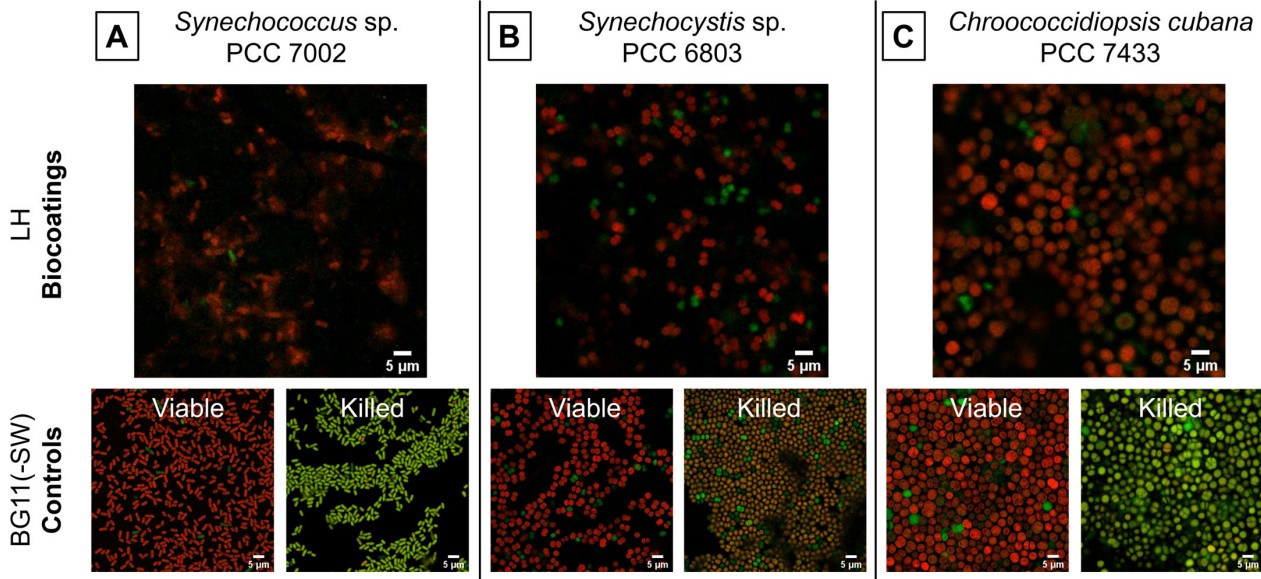

**FIG 5** The cyanobacterial strains retain photosynthetic pigments in the biocoatings. Visualization of the cyanobacteria *Synechococcus* sp. PCC 7002 (A), *Synechocystis* sp. PCC 6803 (B), and *Chroococcidiopsis cubana* PCC 7433 (C) by means of their pigments using CLSM. Chlorophyll *a* fluoresces red, whereas the degradation of chlorophyll *a* reveals the secondary pigments fluorescing green. Bacterial cultures were either visualized untreated (viable controls) or heat-killed (killed controls). Biocoatings containing halloysite (LH) were visualized immediately after rehydration. Most of the bacterial cells within the biocoatings appeared red, which means that they contained chlorophyll *a,* as did the viable controls.

assessed using CLSM but cannot be used as a strict quantitative measure of the viability. Therefore, viability was measured using assays based on ATP and oxygen evolution.

## Determination of bacterial cell viability using ATP

Cell viability within biocoatings was also investigated using the CellTiter-Glo 3D Cell Viability Assay. This endpoint assay lyses cells releasing their ATP which is ubiquitously present in living cells and, therefore, used to assess viability (39, 40). The ATP reacts with the Ultra-Glo rLuciferase from the reagent, resulting in a luminescent signal [relative luminescence units (RLU)] (41). Using standard curves of ATP and cell cultures (Supplemental Material), the ATP levels of the liquid controls, dried bacteria samples, and the biocoatings were estimated.

The dried *Synechococcus* 7002 exhibited low RLU and, therefore, low ATP levels (Fig. 6A), which is indicative of low viability following desiccation, in all suspension types. However, with *Synechocystis* 6803, higher ATP values were observed, indicative of increased survival (Fig. 6B). Statistically significant differences in viability were noted between dried LW and BG11 samples, as well as LW and latex samples (**$p$= 0.0081 and *$p$= 0.0340, respectively), indicating that *Synechocystis* 6803 survives best in the LW mixture. Preliminary experiments showed that the desiccation-resistant *Chroococcidiopsis* 7433 which was dried for 1 month could recover viability after 1 week on fresh BG11 agar (Fig. S4). This was supported by these ATP assays showing that the extremophile *Chroococcidiopsis* 7433 dried in BG11 displayed the highest viability, as evidenced by ATP concentration $\log_{10}(1.34)$ nM (21.68 nM), which was similar to the levels observed from liquid samples at $10^7$ CFU/mL (Fig. 6C). There was statistically significantly lower viability in the undiluted latex compared to the working suspension of LH (****$p < 0.0001$), LW (***$p = 0.0002$), and BG11 (****$p < 0.0001$). Interestingly, *Chroococcidiopsis* 7433 had higher viability in the presence of halloysite (LH) than without halloysite, indicating that this key inorganic mineral component has beneficial effects for the survival of these rock-dwelling bacteria.

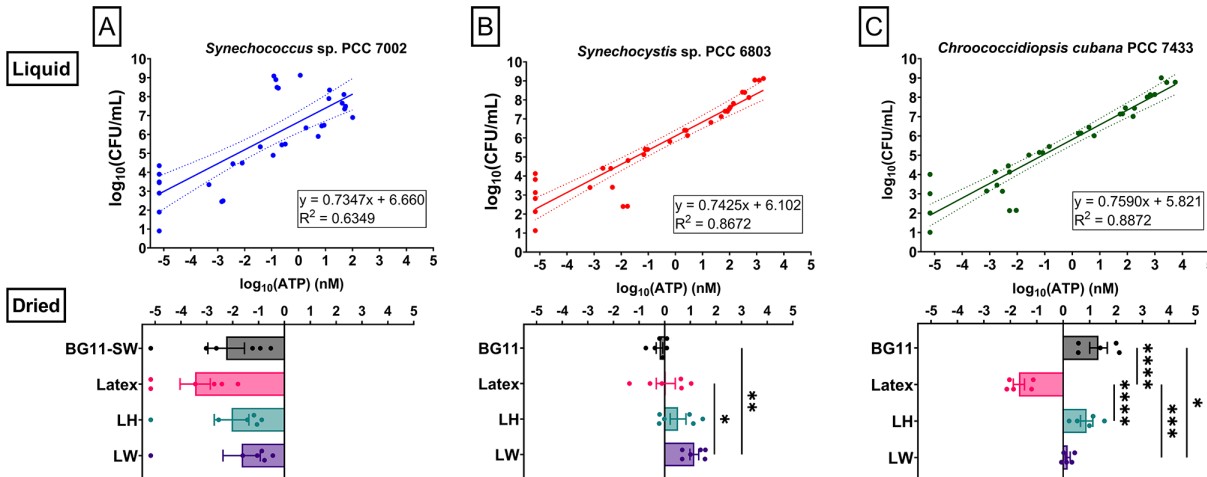

**FIG 6** *Synechocystis* 6803 and *Chroococcidiopsis* 7433 survive desiccation. Estimated ATP concentration in (A) *Synechococcus* sp. PCC 7002, (B) *Synechocystis* sp. PCC 6803, and (C) *Chroococcidiopsis cubana* PCC 7433. Upper panels: the regression of ATP concentration with serial dilutions of liquid samples. Lower panels: estimated ATP concentrations in dried samples, either in medium only [BG11(-SW)] or as biocoatings in latex, latex + halloysite (LH), and latex + water (LW). Mean ± SEM. One-way ANOVA followed by Tukey's multiple comparisons. (A) *Synechococcus* sp. PCC 7002 ($n$ = 6 biological replicates), (B) *Synechocystis* sp. PCC 6803 ($n$ = 6 biological replicates, *$p$ = 0.0340 and **$p$ = 0.0081, respectively). and (C) *Chroococcidiopsis cubana* PCC 7433 ($n$ = 5 biological replicates, *$p$ = 0.0138, ***$p$ = 0.0002, and ****$p$ < 0.0001).

## Measuring the oxygen evolution of the cyanobacterial biocoatings

The biocoatings with *Synechococcus* 7002, *Synechocystis* 6803, and *Chroococcidiopsis* 7433 contained in the range from 10 log CFU/m$^2$ to 11 log CFU/m$^2$ (for biocoatings with thicknesses between 50 and 70 µm) at time $t$ = 0 and were rehydrated by immersion in BG11(-SW). The *Chroococcidiopsis* 7433 biocoatings produced more than 20 mg/L during the light phases due to photosynthesis (Fig. 7A). Biocoatings containing *Synechococcus* 7002 or *Synechocystis* 6803 did not evolve oxygen and were similar to the negative control. Interestingly, in *Synechocystis* biocoatings, oxygen consumption could be observed until Day 5, suggesting that cellular respiration, but not photosynthesis, was occurring. It can be observed that the rates of increase in the dissolved oxygen content were very similar between the three replicates for the *Chroococcidiopsis* 7433 biocoatings. However, there is experimental variability in the concentration of the saturation value in the plateau region, which is set by an equilibrium with the oxygen in the liquid phase, as is given by Henry's law (42). Although each sample was prepared in an identical way (see the Materials and Methods), there might have been some variability in the Parafilm seals on the sample dishes.

Averaging the oxygen production during the light phases allowed us to observe a steady increase in oxygen production from the *Chroococcidiopsis* 7433 biocoatings (Fig. 7B). This oxygen production was significantly higher than from the biocoatings containing *Synechococcus* 7002 and *Synechocystis* 6803 and the negative control, resulting in statistical differences from D1 (vs 7002: **$p$ = 0.0087, vs 6803: **$p$ = 0.0082, and vs negative control: *$p$ = 0.0104), which increased to even higher statistical differences (****$p$ < 0.0001) on D2 (vs 6803) and D4 (vs 7002 and vs negative control).

For *Chroococcidiopsis* 7433 biocoatings, the gradient was measured from the beginning of the light phase to calculate the specific rate of oxygen production as the concentration approached its saturation value (see Fig. S14). This gradient steadily increased over the 6 days of measurements, reaching maximum levels of 0.4 ± 0.1 gO$_2$ g$_{DW}^{-1}$ day$^{-1}$ (Fig. 7C). These results determine that *Chroococcidiopsis* 7433 survived the film formation process and produced oxygen while immobilized in the biocoating.

To determine whether the bacteria remained contained within the biocoatings, the number of bacteria that escaped into the rehydration medium was calculated by both

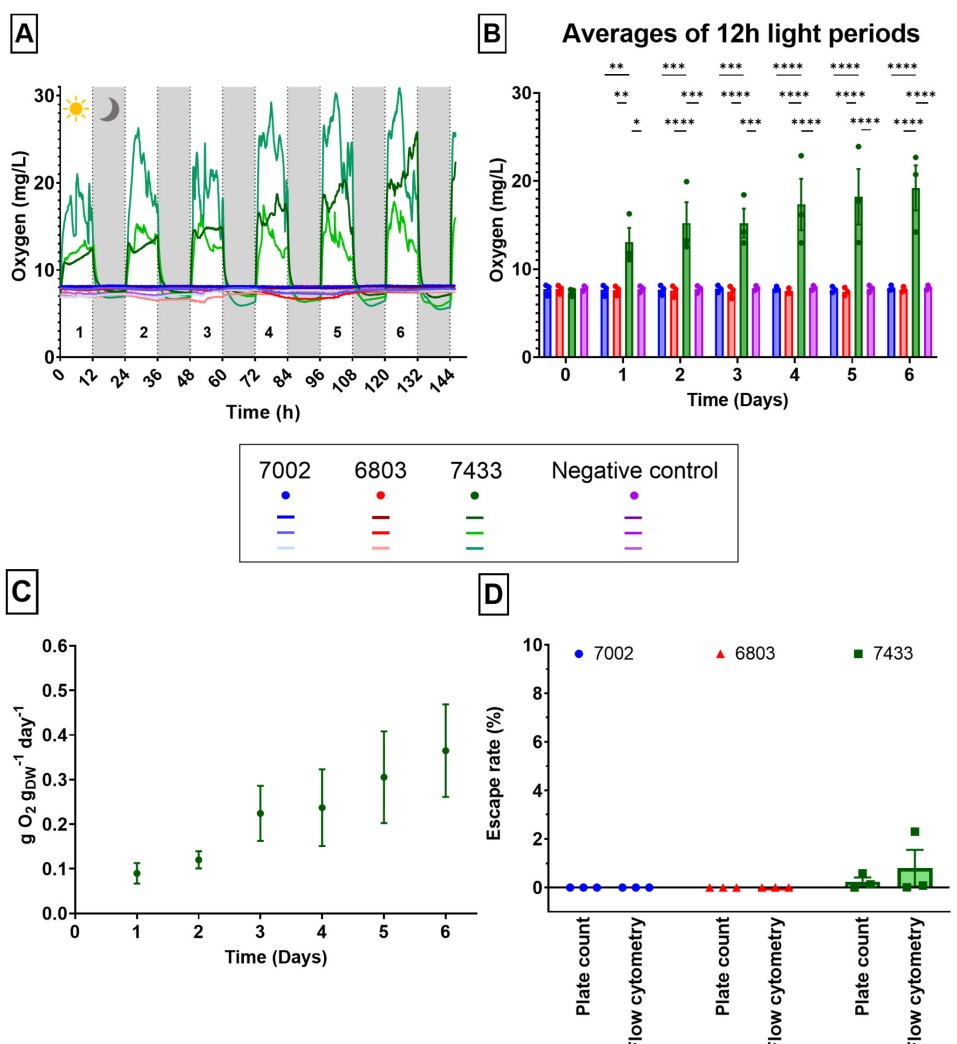

**FIG 7** Biocoatings with *Chroococcidiopsis* 7433 showed oxygen evolution over 6 days with low levels of escape from biocoatings. The oxygen evolution of biocoatings with halloysite (LH) containing *Synechococcus* 7002, *Synechocystis* 6803, or *Chroococcidiopsis* 7433 and abiotic biocoatings (negative control). Three biological replicates are presented with varying shades of the same color. (A) The full data show that the levels of dissolved oxygen varied over the 12 h light periods. *Chroococcidiopsis* 7433 biocoatings were producing oxygen continuously, whereas *Synechocystis* 6803 biocoatings were consuming it. (B) Averages of the 12 h light periods show the trend of *Chroococcidiopsis* 7433 steadily increasing their oxygen production over time. Two-way ANOVA with Tukey's multiple comparisons (Mean ± SEM, **$p = 0.0087$ and $p = 0.0082$, *$p = 0.0104$, ***$p = 0.0001$, and ****$p < 0.0001$ from the upper left down to lower right). (C) The specific rate of oxygen production of *Chroococcidiopsis* 7433 biocoatings during the beginning of the light phases (in mass of oxygen per dry weight per day). The gradient steadily increases over the 6 days (Mean ± SEM). (D) Percentage of bacteria that escaped from the LH biocoatings into the rehydration medium after 6 days of oxygen evolution experiment, measured by plate count method and flow cytometry. *Synechococcus* 7002 and *Synechocystis* 6803 biocoatings did not show any release of viable cells by both methods, whereas *Chroococcidiopsis* 7433 showed a maximum of 0.24% ± 0.18% or 0.80% ± 0.75% escaped cells (plate count method and flow cytometry, respectively) ($n = 3$ biological replicates, Mean ± SEM).

the plate count method and flow cytometry (Fig. 7D). No viable *Synechococcus* 7002 could be recovered from the rehydration medium either by plate count method or by flow cytometry. In the case of *Synechocystis* 6803 biocoatings, only one replicate contained a small number of viable bacteria [~2 log(cells/mL)]. The rehydration liquid from *Chroococcidiopsis* 7433 biocoatings contained ~4–5 log(cells/mL) of viable bacteria, representing 0.24% ± 0.18% (plate count method) or 0.80% ± 0.75% (flow cytometry).

Overall, there was little escape of bacteria, and therefore, the observed photosynthesis and oxygen production resulted from the encapsulated bacteria. It should be noted that these bacteria are environmentally ubiquitous strains.

As the longevity of the harnessed bacteria is an important characteristic for many applications, *Chroococcidiopsis* 7433 was selected, as it survived the film formation process and was functional within the biocoatings for the duration of the initial 7-day time course. Oxygen production was measured over 1 month with three independent *Chroococcidiopsis* 7433 biocoatings and a negative control (Fig. 8). These biocoatings contained an average number (±SD) of $10.20 \pm 0.07$ log CFU/m$^2$. The activity increased on successive days and the media change on D12 did not cause a decrease, which further confirms that the bacteria responsible for the oxygen production were immobilized in the biocoating (Fig. 8A). Furthermore, replenishing the source of carbon (as NaHCO$_3$) did not cause a spike in the oxygen production rate, but the dissolved oxygen concentration was maintained at an approximately constant value, suggesting that the depletion of carbon in the medium over time was not detrimental to the oxygen evolution. Oxygen was still being produced on D27 when the experiment was stopped due to time and space constraints (Fig. 8B). This duration is a lower limit for the lifetime of the biocoatings, provided that the medium is changed or replenished at least once. The specific rate of oxygen production reached up to $0.40 \pm 0.03$ gO$_2$ g$_{DW}^{-1}$ day$^{-1}$ (D23). The rate dropped after the medium change ($0.24 \pm 0.22$ gO$_2$ g$_{DW}^{-1}$ day$^{-1}$, D13) but increased rapidly again after a few days (Fig. 8C). Preliminary oxygen measurements using liquid bacterial cultures, biocoatings, and dried bacterial cultures (dried in the same way as the biocoatings), containing the same number of bacteria at the beginning of the experiment, were performed (Fig. S11). Indeed, there is an initial lag in the oxygen production of the biocoatings compared to the liquid culture. The specific rate of oxygen evolution was approximately two times lower in biocoatings than in the liquid cultures.

Using plate count methods and flow cytometry, it was again determined that the escape of bacilli from the biocoatings was limited to less than 0.19% (Plate count) or 0.08% (Flow cytometry) on D12. This increased to $0.80\% \pm 0.74\%$ and $2.28\% \pm 2.26\%$ (plate count and flow cytometry, respectively) on D27, which is due to partial delamination of one coating (replicate 2) (Fig. 8E).

## DISCUSSION

We have shown the use and application of extremophilic, desiccation-resistant cyanobacteria in a biocoating capable of carbon capture, measured via oxygen evolution. This is the first study of this type to our knowledge. Our non-toxic biocoating suspensions with halloysite (LH) used a latex binder with a higher hardness and offered greater mechanical protection under ambient conditions. Therefore, our biocoatings could be submerged in the rehydration liquid, rather than being rehydrated by capillary force as in previous studies (2, 18, 43, 44). This is an advantage as the biocoatings could be even applied to non-porous substrates. Hard biocoatings would allow their handling and transport without damage.

Imaging by SEM revealed the complex structure of the halloysite and latex and the morphological differences of the bacteria. EDX is already widely used in other fields, but only starting to find applications in microbiology, for example, to detect food pathogens (via C, O, and N) or to detect accumulated calcium carbonate in cyanobacteria (27, 45, 46). This is the first time EDX has been used in the context of biocoatings and is an ingenious method to identify components and the structural complexity of biocoatings not only by shape but by their elemental composition. This is of importance in situations when the size and shape of the components are similar.

CLSM using the light-harvesting pigments helped estimate the distribution of the bacteria within the biocoatings. The use of light-harvesting pigments alone indicated the presence of chlorophyll *a* and phycobilisomes of all strains in the biocoatings. However, viability assays using ATP showed *Synechococcus* 7002 did not survive, while *Synechocystis* 6803 and *Chroococcidiopsis* 7433 survived the film formation. Following

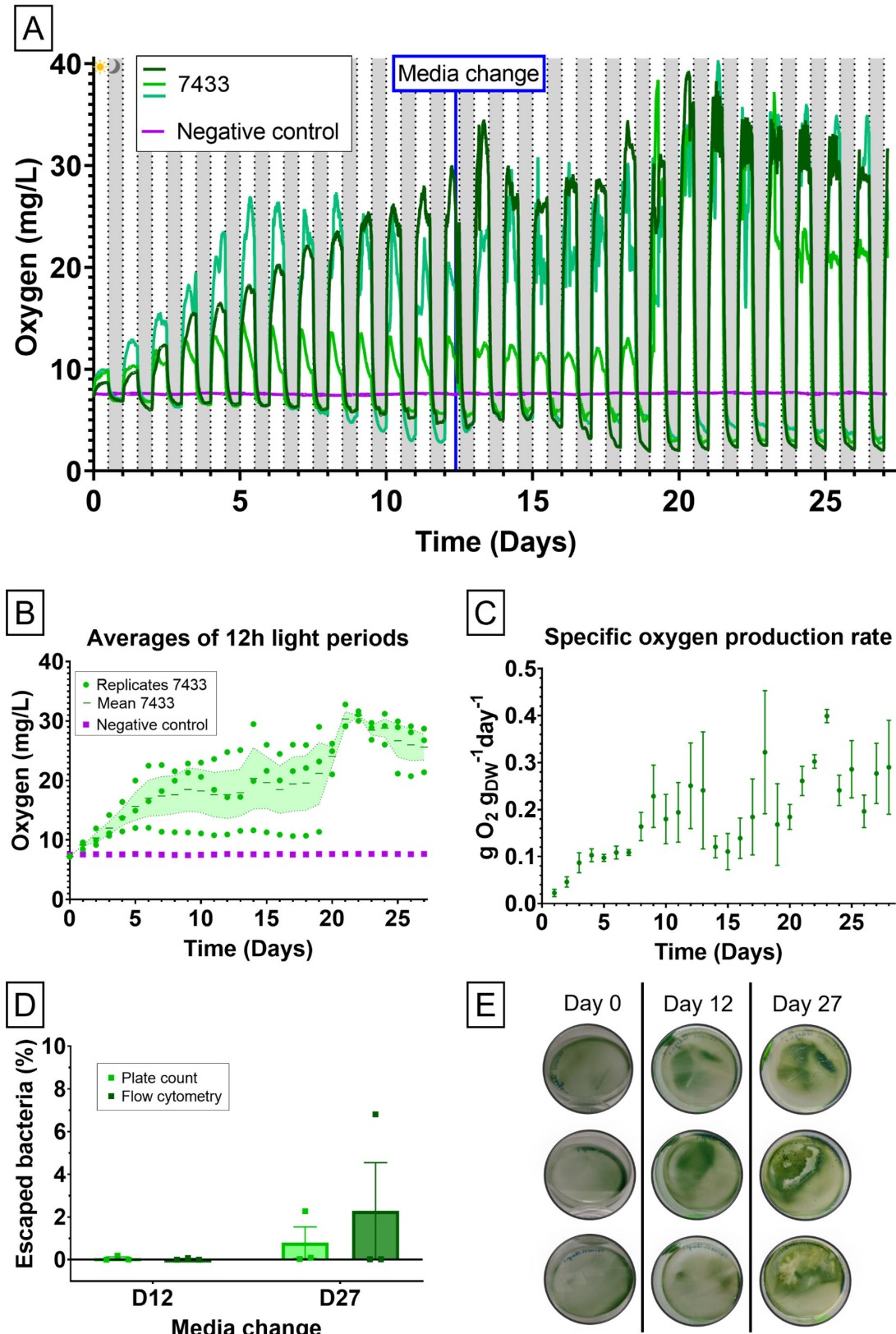

**FIG 8** *Chroococcidiopsis* 7433 was able to produce oxygen within biocoatings for 1 month and negligible amounts of cells escaped the biocoatings. Long-term oxygen evolution of LH biocoatings containing *Chroococcidiopsis* 7433 (three biological replicates) and an abiotic coating (negative control). The experiment ran for 1 month and featured a BG11 medium change on Day 12. (A) Full data of the oxygen evolution during the light and dark phases. The oxygen evolution

**FIG 8** (Continued)

was observed during the light phases, and oxygen consumption was observed during the dark phases. (B) Averages of the oxygen evolution during 12 h light periods with the three biological replicates (points), their mean (dash), and standard error of the mean (dotted line). It shows the overall trend of increased oxygen production. The medium change on Day 12 did not result in a drop in oxygen production. (C) Specific oxygen production rate of *Chroococcidiopsis* 7433 biocoatings over 28 light phases. In relation to the dry weight, the oxygen production slowed down after the medium change (after 13th light phase) but increased quickly again (Mean ± SEM). (D) Percentage of bacteria that escaped from the LH biocoatings containing *Chroococcidiopsis* 7433 into the rehydration medium during the long-term oxygen evolution experiment, measured by the plate count method and flow cytometry. Low numbers of escaped bacteria were measured on D12, whereas one replicate had higher escapes on D27 because of damage to the biocoating (0.80% ± 0.74% and 2.28% ± 2.26% by the plate count method and flow cytometry, respectively) ($n$ = 3 biological replicates, Mean ± SEM). (E) Photographs of three replicate biocoatings in Petri dishes on Days 0, 12, and 27.

on from this, of these two strains, only the desiccation-resistant *Chroococcidiopsis* 7433 continued to produce oxygen, meaning they were able to carry out photosynthesis. One explanation for the enhanced survival in desiccating environments could be the production of sucrose and trehalose by the bacteria, which have also been accumulated by *Chroococcidiopsis* spp (47, 48). An oxygen experiment over a longer timeline was subsequently designed using *Chroococcidiopsis* 7433. The oxygen production increased from one light phase to the next until reaching a plateau on D6. The first medium change took place on D12, revealing that, once replaced, the oxygen production did not return to its initial values from D0 but continued where it had left off on that day. This suggests that the oxygen is produced by the immobilized bacteria and not by the free-floating escapees.

In past studies, biocoatings were placed in airtight bottles with a defined 5% $CO_2$-enriched gas mixture whose headspace was measured with a $CO_2$ meter every 2 days (2, 43, 44) or flushed with a gas mixture of 80% $N_2$ and 20% $CO_2$ which was analyzed by gas chromatography (18). In this study, the oxygen production was measured continuously in a loosely closed container every 2 min as dissolved oxygen in the rehydration medium above the biocoatings. As the dissolved oxygen content rose in the medium, it equilibrated with the vapor phase to approach an equilibrium defined by Henry's law (42).

Compared to the previously described methods, our method bears the disadvantage of not measuring the total oxygen production per day, as the dissolved oxygen was not removed after each measurement. The specific oxygen production rate was measured when the light phase began (49–51) (up to 0.4 $gO_2$ $g_{DW}^{-1}$ $day^{-1}$). The oxygen evolution and the carbon capture are related mathematically via the photosynthetic quotient (PQ). By using the PQ value of 1.3 reported in other research (18, 52), we estimated the $CO_2$ capture to be 0.31 $gCO_2$ $g_{DW}^{-1}$ $day^{-1}$, which is in the same range as found for other confined cyanobacteria (2). Although the cyanobacteria would probably not be able to continue capturing carbon dioxide and producing oxygen at this rate over the whole light phase because of photorespiration and the production of reactive oxygen species (53, 54), analysis of the early phase allowed an estimate of the specific oxygen production rate.

## Conclusions

Using cyanobacteria in biocoatings for biotechnological applications is problematic as the bacteria do not always survive the processing. However, we have shown for the first time that the extremophile *Chroococcidiopsis cubana* PCC 7433 can survive the range of stresses and evolve oxygen with a specific rate as high as 0.3–0.4 $gO_2$ $g_{DW}^{-1}$ $day^{-1}$ over the period of 1 month. There is enormous potential in carbon-capturing applications for this extremophilic biocoating, e.g., in space stations or for the production of bioethanol. This strain is also a potential candidate for biotechnology applications in extreme environments. Future work should concentrate on optimizing the use of this strain in such applications and, indeed, the use of extremophiles in biocoatings, in general.

## MATERIALS AND METHODS

### Culturing the cyanobacteria

Axenic cultures of cyanobacteria *Synechococcus* sp. PCC 7002 (*Synechococcus* 7002), *Synechocystis* sp. PCC 6803 (*Synechocystis* 6803), and *Chroococcidiopsis cubana* PCC 7433 (*Chroococcidiopsis* 7433) were acquired from the Pasteur Culture Collection of Cyanobacteria (PCC) (Paris, France). Fifty times concentrated BG11 (blue-green) growth medium (Sigma-Aldrich, C3061, Gillingham, UK) was diluted to 1× for liquid cultures using Milli-Q water before autoclaving at 121°C for 15 min. Five millimolar of $NaHCO_3$ was added immediately prior to use. To create BG11-SW (seawater variant) for *Synechococcus* 7002, NaCl at 10 g/L (Merck, 567440, Gillingham, UK) and vitamin B12 at 10 µg/L (Sigma, V2876) were added. All strains were grown while shaking at 90 rpm at 25°C under a cycle of 12 h light (~1,200 lux) and 12 h dark.

Liquid cultures were used for experiments or subculture after an incubation period of 21–35 days. The bacterial numbers were estimated by optical density (OD) at 730 nm ($OD_{730}$) (Thermo Scientific GENESYS 30 Vis Spectrophotometer, Horsham, UK) and colony-forming units per milliliter (CFU/mL). The plate count (CFU/mL) method was carried out by making serial 10-fold dilutions, followed by placing 10 µL drops in triplicate (Miles and Misra) on solid culture media (35). Solid media consisted of 2 × BG11 and 0.6%–0.75% (wt/vol) Bacto Dehydrated Agar (BD, 214010, Plymouth, UK) supplemented like the liquid cultures. CFUs were determined after 14 days of incubation at 25°C under ~1,500 lux and recalculated to CFU/mL using the appropriate dilution factor.

### Biocoating preparation

The sterile synthetic latex was a copolymer of methyl methacrylate ($C_5H_8O_2$), *n*-butyl acrylate ($C_7H_{12}O_2$), and methacrylic acid ($C_4H_6O_2$) (in a molar ratio of 56:41:3 ($T_g$ = 34°C) as previously described (16). The pH was adjusted to 7 by adding 1 M NaOH. The hardness of dried 100 µm thick coatings on glass plates was measured via a König pendulum hardness test (Sheen Instruments, Cambridge, UK) at 20°C in the angular range from 6° to 3°. As a comparator, a low-$T_g$ latex ($T_g$ = 20°C) consisting of the same copolymer, but in a molar ratio of 50:47:3, was used.

An aqueous suspension of the clay mineral, halloysite, was prepared in Milli-Q water with a final concentration of 15 wt.% halloysite [$Al_2Si_2O_5(OH)_4 \cdot 2H_2O$, Sigma-Aldrich, 685445] and stabilized with 5 wt.% PEG [$H(OCH_2CH_2)nOH$, Sigma-Aldrich, 76293] and autoclaved (121°C for 15 min).

The latex and halloysite (LH) suspension was prepared by adding the latex dispersion to the halloysite suspension in a 1:2 vol ratio and mixing it with a vortex. The pH was adjusted to 7 and sonicated in an ice bath for 10 min to achieve a fine suspension. A control suspension consisting of latex and water (LW) was prepared in the same way. Two more control suspensions, undiluted latex (Latex) and BG11 medium (BG11), were also used.

Bacteria were grown to stationary phase ($10^8$–$10^9$ CFU/mL) and harvested by centrifugation at 5,000 × *g* at room temperature (RT) (Thermo Scientific, Megafuge 16R) prior to adding to the suspensions RT.

For toxicity and viability assay experiments, biocoatings were prepared by mixing 75 µL of concentrated bacteria ($10^9$–$10^{10}$ CFU/mL) with 250 µL of suspension. For oxygen evolution experiments, 600 µL of bacteria and 2 mL of LH were used.

For film formation, the volume of the suspension was calculated to achieve a dry thickness of 50–70 µm. Biocoatings were film-formed at 37°C for 6–7 h until visibly dry. During the process, the samples were covered by the respective lid to create a high-humidity atmosphere and suppress water evaporation for 1 h. Thereafter, the containers were left partially open for 5–6 h, while switching the open side every 1.5 h to ensure uniform evaporation across the entire dish or plate, until the biocoatings were visibly dry and rehydrated.

## Toxicity testing of the wet latex

The toxicity of the respective suspensions (LH, LW, Latex, and BG11) was assessed immediately upon mixing with the bacteria [Day (D) 0] and after storing the suspension at RT in the dark for 7 days without any film formation. The plate count method was used to determine the CFU/mL.

## Scanning electron microscopy and Energy-dispersive X-ray spectroscopy

Individual components of the biocoatings, as well as film-formed and cross-sectioned biocoatings (see Supplemental Material), were visualized and characterized by SEM/EDX using JEOL JSM-7100F Thermal Field Emission SEM.

The samples were mounted onto a nut and SEM stub using carbon tape and sputter-coated with a 9-nm gold layer (Quorum Q 150 ES plus, Laughton, UK). SEM was performed using a low accelerating voltage (5 kV) and low probe current (6 pA), while EDX required a higher probe current (11 pA). The data analysis was performed using the Pathfinder software 2.5 (Thermo Fisher). Several points were characterized in each sample, but only up to three representative points are presented for clarity.

## Confocal laser scanning microscopy

Biocoatings were visualized by plan view, with live bacterial culture used as the live control and heat-killed bacteria (96°C for 30 min) as the dead control. Both were imaged immediately after rehydration.

Images were obtained using a Nikon A1M Confocal Microscope (Nikon A1M on Eclipse Ti-E, Amstelveen, The Netherlands) with a 60 × Plan Apochromat λ oil immersion objective (NA 1.4, WD 0.13). Chlorophyll *a* and phycobilisomes were visualized using the excitation wavelength of 561 nm and a 595/50-nm emission filter cube. The non-specific green fluorescence was excited at 488 nm, and the emitted fluorescence was filtered by a 525/50-nm emission filter cube. Analysis was performed using ImageJ 1.53k software [Fiji, (55, 56)].

## Bacterial cell viability assay (via ATP concentration)

The CellTiter-Glo 3D Cell Viability Assay (Promega, G9681, Madison, WI, USA) was used to measure the ATP concentration of biocoatings on D0 and D7 (following dry storage at RT in the dark). Triplicates of samples of 10 µL of the biocoating suspension were placed into white 96-well plates (Thermo Scientific Sterilin, 611F96WT), film-formed, and rehydrated with 100 µL BG11(-SW). The assay was carried out following the manufacturer's protocol and the luminescence read on the CLARIOstar *Plus* (BMG LABTECH, Aylesbury, UK) (Supplemental Material).

## Oxygen evolution measurements

Two milliliters of LH biocoatings and abiotic coatings was spread with Cell Scrapers (Fisherbrand, 11597692, Loughborough, UK) on 90-mm round polystyrene Petri dishes (Fig. 9A). Following complete film formation and rehydration with 25 mL BG11(-SW), a fiber-optic oxygen probe (Pyroscience, Aachen, Germany) was placed in the liquid of each dish before wrapping Parafilm around the edges to seal the dishes lightly. The dissolved oxygen was recorded every 2 min under light/dark cycles of 12 h each [FireSting-PRO (4 Channels), Pyroscience] (see Fig. 9B and photographs in Fig. S8 and S9). The conditions approximated a closed system such that the oxygen in the liquid phase became saturated when reaching equilibrium with the vapor.

The specific rate of oxygen production was measured from the gradient of a linear plot of the mass of oxygen (~20 mL) against time during the beginning of the light phase (Fig. S14). The rate was calculated per the dry weight (DW) of the bacteria per day ($gO_2$ $g_{DW}^{-1}$ $day^{-1}$) (in triplicates). The dry weight of the bacteria was estimated on Day 0 of the experiments. One milliliter of bacterial cultures in the stationary phase was placed in

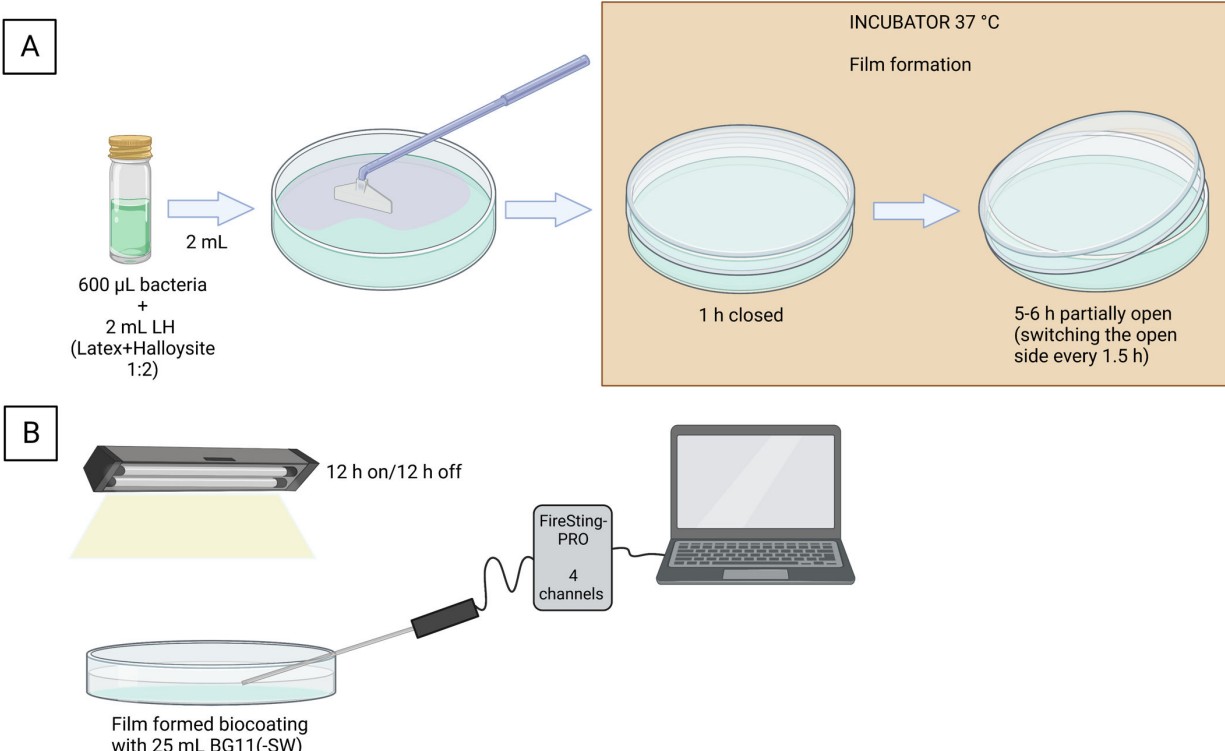

**FIG 9** Biocoating preparation method for oxygen evolution measurements. (A) 600 µL bacteria and 2 mL LH (latex + halloysite 1:2) were mixed together, and from this mixture, 2 mL were spread on a Petri dish using a cell scraper. The biocoatings were incubated at 37°C to allow film formation. The Petri dish was closed using the lid for 1 h, followed by 5–6 h of leaving the dish partially open, switching the open side every 1.5 h. (B) Following the film formation, the biocoating was rehydrated using 25 mL of BG11(-SW). To measure oxygen evolution, a fiber-optic probe, connected to the FireSting-PRO and a laptop, was placed in the Petri dish. The Petri dish and its lid were closed and sealed using Parafilm. Illumination using a LED lamp above the Petri dish followed on/off cycles of 12 h. Images created with Biorender.com.

Eppendorf tubes and weighed before and after evaporation at 97°C for 5 h. The mean value of the dry weight of 3–8 independent cultures was used to calculate the oxygen production per dry weight of all time points.

The rehydration medium was collected from the biocoatings, centrifuged at 5,000 × $g$ at RT for 10 min, and redispersed in 1 mL. This volume was used for the plate count method and flow cytometry. Samples for flow cytometry were fixed in 1% paraformaldehyde and processed within 1 week. From the ~8 log cells immobilized on the Petri dish surface, at most ~7 $\log_{10}$(cells/mL) were isolated in the rehydration medium.

## Flow cytometry

Flow cytometry using the Thermo Fisher Attune Acoustic Focussing Flow Cytometer was performed on fixed rehydrated liquid samples to estimate the number of viable and dead cells. Just before use, 10 µL of counting beads CountBright Absolute Counting Beads (Thermo Fisher, C36950) was added to 100 µL of samples and vortexed. No stain was added, as the settings established in-house on cultures (live and killed cells) allowed differentiation between the live (red) and dead (green) cyanobacteria via their pigments (Supplemental Material). All statistical analyses were performed using GraphPad Prism 9.5 (for Windows, www.graphpad.com).

## ACKNOWLEDGMENTS

This research was funded by a Research Project Grant from The Leverhulme Trust. The funders had no role in study design, data collection and interpretation, or the decision to submit the work for publication.

We thank Stefan A. F. Bon and Joshua R. Booth (University of Warwick) for the synthesis of the latices. We thank Jana Baron, Jim Barber, Suraj Songra, Jessica Kingshott, Miles Choularton, Gill Wallis, Jenny Spinks, Vasiliki Tsioligka, Shaaezmeen Basheer, and David Jones (University of Surrey) for technical assistance.

S.K.: conceptualization (lead); data curation (lead); formal analysis (lead); investigation (lead); methodology (lead), writing—original draft preparation (lead). Y.C.: methodology (supporting); resources (supporting); writing—review and editing (supporting). S.H.-W.: review and editing (equal); supervision (equal); writing—review and editing (equal), conceptualization (equal), funding acquisition (equal). J.L.K.: review and editing (equal); supervision (equal); conceptualization (equal); funding acquisition (equal); project administration (lead), writing—review and editing (equal).

The authors declare no competing interests or conflict of interest.

## AUTHOR AFFILIATIONS

[1]Department of Microbial Sciences, School of Biosciences, University of Surrey, Guildford, Surrey, United Kingdom

[2]School of Mathematics and Physics, University of Surrey, Guildford, Surrey, United Kingdom

## PRESENT ADDRESS

Yuxiu Chen, School of Engineering, University of Newcastle, Newcastle upon Tyne, United Kingdom

## AUTHOR ORCIDs

Simone Krings http://orcid.org/0000-0002-9482-8372
Yuxiu Chen http://orcid.org/0009-0001-2218-1055
Joseph L. Keddie http://orcid.org/0000-0001-9123-183X
Suzanne Hingley-Wilson http://orcid.org/0000-0002-6514-1424

## FUNDING

| Funder | Grant(s) | Author(s) |
|---|---|---|
| Leverhulme Trust | RPG-2018-393 | Simone Krings |
| | | Yuxiu Chen |
| | | Joseph L. Keddie |
| | | Suzanne Hingley-Wilson |

## AUTHOR CONTRIBUTIONS

Simone Krings, Conceptualization, Data curation, Formal analysis, Investigation, Methodology, Writing – original draft | Yuxiu Chen, Methodology, Resources, Writing – review and editing | Joseph L. Keddie, Conceptualization, Funding acquisition, Project administration, Supervision, Writing – review and editing | Suzanne Hingley-Wilson, Conceptualization, Funding acquisition, Supervision, Writing – review and editing

## DATA AVAILABILITY

All data are available on figshare (https://doi.org/10.6084/m9.figshare.23284340.v1 and https://doi.org/10.6084/m9.figshare.23284337.v1).

## ADDITIONAL FILES

The following material is available online.

### Supplemental Material

**Supplemental figures (Spectrum01870-23-s0001.docx).** Fig. S1 to S15.

### Open Peer Review

**PEER REVIEW HISTORY (review-history.pdf).** An accounting of the reviewer comments and feedback.

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
