## [Reviewer comments · Microbiology Spectrum]

Microbiology Spectrum

Oxygen evolution from extremophilic cyanobacteria confined in hard biocoatings

Simone Krings, Yuxiu Chen, Joseph Keddie, and Suzanne Hingley-Wilson

Corresponding Author(s): Suzanne Hingley-Wilson, University of Surrey

Review Timeline:

Submission Date:	May 5, 2023
Editorial Decision:	June 16, 2023
Revision Received:	July 10, 2023
Editorial Decision:	July 26, 2023
Revision Received:	August 2, 2023
Accepted:	August 4, 2023

Editor: Ilana Kolodkin-Gal

Reviewer(s): The reviewers have opted to remain anonymous.

Transaction Report:

DOI: <https://doi.org/10.1128/spectrum.01870-23>

June 16, 2023

Dr. Suzanne Hingley-Wilson
University of Surrey
Microbial and cellular Sciences
Stag Hill campus
Guildford, Surrey GU3 1DY
United Kingdom

Re: Spectrum01870-23 (Oxygen evolution from extremophilic cyanobacteria confined in hard biocoatings)

Dear Dr. Suzanne Hingley-Wilson:

Link Not Available

Sincerely,

Ilana Kolodkin-Gal

Journals Department
Reviewer comments:

Reviewer #1 (Comments for the Author):

Several interesting observations and discoveries are reported in this paper. Especially the successful use of halloysite as a latex binder. And the demonstration of oxygen production by *Chroococcidiopsis* immobilized in latex. The paper was a bit difficult to read because, except for the very last part of the results, equal emphasis is given to all data including quite a bit of negative data. This diminished the impact of the paper substantially. I understand that one wants all their data to be included but oftentimes less is more. I urge that the paper be substantially revised with this in mind. I give some specific examples below.

1. Questions I had throughout are:

- the biocoatings with cells embedded in them were hard - but how hard? Were they at all pliable or were they stiff? I can't get a sense of this from section 1 of the results.
- How did you fabricate the biocoatings so they were of uniform thickness?
- Once the biocoatings were fabricated and dried, it seems that they subsequently placed into liquid. Was everything done as described in lines 519-523? If not, more description for other situations is needed. The type of rehydration liquid used should be stated in the body of the paper.
- I read the methods carefully but could not quite visualize the set-up.

2. Instead of referring to *Chroococcidiopsis* as an extremophile in the abstract it might make more sense to indicate that it is desiccation resistant. Has desiccation resistance been tested for this strain? It should be. It would strengthen the paper to show that it is extremely desiccation resistant. Or at least do some investigation into why it does well in the biocoatings relative to the other strains.

3. pgs.10 and 13, SEM, EDX and CLSM should be spelled out when used for the first time.

4. p. 10 "Visualization of biofilm coating." I did not find this section to be useful or informative in any way. In general Figure 4 detracts from the overall message of the paper and would be better published elsewhere. I agree that EDX may be a useful tool to identify the different components of biocoatings, but the paper's conclusions don't in any way depend on these data.

5. The statement in the abstract that PCC 7002 is nonviable in the coatings along with data in Fig 6, showing low ATP levels, is not consistent with the image in Fig 5A, which based on chlorophyll a fluorescence indicates that the cells are viable in coatings. One of the two assays is not accurately reflecting viability.

6. p. 16 line 286: does 10-11 log CFU mean 10¹⁰-11? And if it does, then this seems like a very low number of cells over a m². Moreover, the thickness of the coatings should be mentioned here because a m² isn't a volume.

7. p. 16 lines 289-294. I don't understand this statement. If an experiment was not executed properly, it should be redone - or mention of it removed. Best to remove these sentences.

8. Fig 7 could be removed from the MS. Once you state that strains 7002 (which is dead) and 6803 do not produce oxygen, they should not appear again in the data shown in Fig. 7. The negative control is sufficient. The focus now should be solely on 7433 and all relevant data are presented in Fig. 8

9. How does a specific rate of oxygen evolution of .4 g/g biomass per day by *Chroococcidiopsis cubana* embedded in a biocoatings compared to its rate of oxygen evolution in culture?

10. Discussion. It is important to be more precise about what kind of extremophile *Chroococcidiopsis* is. Extremophile is a broad term.

Reviewer #2 (Comments for the Author):

In brief, the study by Krings et al. describes encapsulation of cyanobacteria inside biocoatings and comparisons of three cyanobacterial species in regards to survival and photosynthetic activity. Cyanobacteria serve as a significant production platform and therefore it is important to seek for new growth modes, which will be further tailored for specific harvesting of biomass or metabolites. The study provides proof-of concept for the thermophilic cyanobacterium *Chroococcidiopsis cubana* PCC 7433 - it describes particular engineering protocol and demonstrates sustained photosynthetic oxygen evaluation. Overall, the study is nicely done and the manuscript is well written. A few comments and suggestions that need to be addressed are listed below.

I am concerned about the interpretation of data presented in Fig. 5. The cited paper shows that heated cells lose their red fluorescence. However, not all lethal stresses cause decreased red cyanobacterial fluorescence and vice versa, decreased red cell fluorescence does not necessarily represent cell death. For example, nutrient limited cells actively degrade their Phycobilisomes, the major light harvesting complexes, which emit red fluorescence. Those starved cells are still viable. Indeed, interpretation of red/green fluorescence as an assessment for viability (Fig. 5) is NOT in accordance with the viability determination using CellTiter-Glo (Fig. 6). The three different cyanobacterial species tested exhibit similar red fluorescence but *Synechococcus* is considered non-viable according to the ATP method. In summary of this part - fluorescence data should not be regarded as an indication for viability. The authors may employ the commonly used live/dead SYTOX staining to support the viability findings by ATP measurements. Statements such as in lines 394-396: "While *Synechococcus* 7002 did not survive, *Synechocystis* 6803 and *Chroococcidiopsis* 7433 survived the film formation, as determined by viability assays based on CLSM and ATP" need to be revised.

Of note, the authors attribute red fluorescence to Chla, which is correct; however, as pointed out above, Phycobilisomes are the predominant pigments under the growth conditions used in this study are the main source of red fluorescence.

Methods indicate that photosynthetic activity was calculated per dry weight (DW) of the bacteria. The cells, however, are embedded within the biocoatings so it is not clear how dry weight of the cells was determined. Please clarify. Is it possible to extract chlorophyll and normalize data to this pigment?

Lines 291-294: "However, the seal was not fully closed..." Does this mean that may have had higher oxygen evolution rate? Was the seal fully closed in other measurements? Why not repeat? Please clarify.

In regards to Figure 4A (f-h): Perhaps it is possible to identify some rod-shaped bacteria in f but I do not see cells in g and h. Use arrow heads to indicate the cells.

All acronyms should be defined first time they appear in text.

Staff Comments:

Preparing Revision Guidelines

Please return the manuscript within 60 days; if you cannot complete the modification within this time period, please contact me. If you do not wish to modify the manuscript and prefer to submit it to another journal, please notify me of your decision immediately so that the manuscript may be formally withdrawn from consideration by Microbiology Spectrum.

Response to Reviewers

We thank the reviewers for their time and for their helpful comments. We addressed them individually below. References to page numbers and lines are made in accordance with the marked-up manuscript.

Reviewer #1 (Comments for the Author):

Several interesting observations and discoveries are reported in this paper. Especially the successful use of halloysite as a latex binder. And the demonstration of oxygen production by *Chroococidiopsis* immobilized in latex. The paper was a bit difficult to read because, except for the very last part of the results, equal emphasis is given to all data including quite a bit of negative data. This diminished the impact of the paper substantially. I understand that one wants all their data to be included but oftentimes less is more. I urge that the paper be substantially revised with this in mind. I give some specific examples below.

We appreciate the care and attention from this reviewer in examining our manuscript. One of the aims of the research is to explore the effectiveness of the three cyanobacterial strains in producing oxygen when confined in coatings. We found that *Chroococidiopsis* was by far the most effective. However, to show that its oxygen evolution was statistically significant compared to the others, we believe it is important to show data from all three species together. The results from the other two species, including the microstructures in Figure 4 and viability data in Figures 5 and 6, might be considered to be “negative data”, but it is still highly valuable in informing the future development of biocoatings. We worry that if the data for the other two species was removed from the manuscript (or moved to the Supplemental Material) it would not be available (or noticed) by other researchers. We appreciate the “less is more” philosophy and have worked to keep the manuscript as brief as possible, putting a good amount of data in the Supplemental Material).

1. Questions I had throughout are:

- the biocoatings with cells embedded in them were hard - but how hard? Were they at all pliable or were they stiff? I can't get a sense of this from section 1 of the results.

The König hardness test, which was used in this research, is a standard test of hardness. However, we appreciate that most readers will not have an intuitive feel for how the numerical value relates to the hardness that one experiences. The hardness of the copolymer used in the coatings was sufficiently hard so that they could not be indented with a thumbnail. By comparison, the coatings made using the lower- T_g copolymer had a lower hardness and could be indented with a thumbnail. Hardness is important for the application of biocoatings, because it correlates with abrasion resistance. A coating with a lower hardness could be abraded or gouged under lower stresses and hence would lack durability in use.

The coating's stiffness (or pliability), as asked about by the reviewer, is measured through the elastic modulus. Our group previously published an article reporting the elastic modulus of the coating materials, both with and without added halloysite. The polymer coating has some brittleness, but the addition of halloysite reduces the brittleness along with the elastic modulus. There is a visual representation of the flexibility of the coatings containing 20 vol.% halloysite in Fig. 7E of that article (See ref. 16: Chen *et al.*, 2020, <https://dx.doi.org/10.1021/acs.biomac.0c00649>). The exact same copolymer was used in this previous research, but the hardness was not reported previously.

We have now added some additional description in Section 1 of the Results. We note that the comparator can be indented with a fingernail. We report the elastic modulus of the copolymer (from our previous research) and state that flexibility is imparted by the addition of the halloysite (page 8, Lines 141-142 and 144-146).

- How did you fabricate the biocoatings so they were of uniform thickness?

The biocoatings were spread evenly with a pipette tip (in 96-well plates) or a cell spreader (in Petri dishes) and then kept horizontally. The film formation was performed starting with one hour of complete covering and then switching sides of the opening in order to allow even drying.

In the future, larger sheets could be coated using an applicator, such as a spiral bar applicator.

To address the reviewer's question, we have now added a new diagram (new Figure 9) with a detailed caption to illustrate the film formation process in the samples for oxygen evolution measurements (see below).

- Once the biocoatings were fabricated and dried, it seems that they subsequently placed into liquid. Was everything done as described in lines 519-523? If not, more description for other situations is needed. The type of rehydration liquid used should be stated in the body of the paper.

This is correct. In the case of the oxygen evolution measurement, BG11 (for 6803 and 7433) and BG11-SW (for 7002) were used as rehydration liquids, as is described on lines 519-523 in the original manuscript.

We have now added a statement at the start of Section 5 to confirm that BG11 was used for rehydration.

- I read the methods carefully but could not quite visualize the set-up.

The production of the biocoatings was described in section "Biocoating preparation" (pages 22 and 23). In response to the comment, a new diagram has now been added to help to illustrate the set-up (new Figure 9). Photographs are also available in the Supplemental Material (Figures S8 and S9 on Supplemental Page 9) to show the even illumination of the biocoatings and the position of the probes. We present it below for

ease of reference. We now state explicitly that photographs are available in the Supplementary Figures 7 and 8 so that reader's attention is drawn to them.

Figure 9. Biocoating preparation method for oxygen evolution measurements. (A) 600 µL bacteria and 2 mL LH (Latex + Halloysite 1:2) were mixed together and from this mixture, 2 mL were spread on a Petri dish using a cell scraper. The biocoatings were incubated at 37 °C to allow for film formation. The Petri dish was closed using the lid for 1 h, followed by 5-6 h of leaving the dish partially open, switching the open side every 1.5 h. (B) Following the film formation, the biocoating was rehydrated using 25 mL of BG11(-SW). To measure oxygen evolution, a fiber-optic probe, connected to the FireSting®-PRO and a laptop, was placed in the Petri dish. The Petri dish and its lid were closed and sealed using Parafilm. Illumination using a LED lamp above the Petri dish followed on/off cycles of 12h. Images created with Biorender.com.

2. Instead of referring to *Chroococidiopsis* as an extremophile in the abstract it might make more sense to indicate that it is desiccation resistant. Has desiccation resistance been tested for this strain? It should be. It would strengthen the paper to show that it is extremely desiccation resistant. Or at least do some investigation into why it does well in the biocoatings relative to the other strains.

A phrase has been added to the abstract to confirm that the bacteria are desiccation-resistant. The desiccation resistance of this genus has been described by several research groups. As an example, Fagliarone *et al.*, 2017 (ref. 26) showed that members of this genus were able to survive air-dried storage for four years. They compared the desiccation resistance to *Synechocystis* sp. PCC 6803 as well. We have now added this example of desiccation-resistance to the Introduction.

We carried out preliminary experiments, which showed that *Chroococidiopsis cubana* PCC 7433 dried on BG11 agar for 1 month could be transferred onto a fresh agar plate. After one week, these bacteria recovered their viability and were able to produce gases. The images below were added to the Supplemental Material as Figure S4 and referred to in the manuscript on page 12, Lines 237-240.

Dried for 1 month 1 week later on fresh BG11

Previous studies showed the production of trehalose and sucrose by members of the *Chroococidiopsis* genus (HersHKovitz *et al.*, 1991 and Fagliarone *et al.*, 2020). However, the focus of this paper was the biocoatings, and the description of the trehalose and/or sucrose production has been described in the past. In response to the reviewer's comment, we have added new text to the third paragraph of the Discussion to highlight the trehalose as an explanation (pages 16-17, lines 338-349).

3. pgs.10 and 13, SEM, EDX and CLSM should be spelled out when used for the first time.

The acronyms have been spelled out when used for the first time.

4. p. 10 "Visualization of biofilm coating." I did not find this section to be useful or informative in any way. In general Figure 4 detracts from the overall message of the paper and would be better published elsewhere. I agree that EDX may be a useful tool to identify the different components of biocoatings, but the paper's conclusions don't in any way depend on these data.

Our aim here was to confirm that the cells were indeed immobilised, and to determine how evenly they were distributed within the biocoatings. For instance, the cells might have been in clusters or fully isolated. The images show that the bacteria were not all sedimented to the bottom of the coatings, as is sometimes the case (see ref. 16). One might argue that to conclude that bacteria in the biocoatings are evolving oxygen, it needs to be established convincingly that the cells are indeed immobilized in the latex polymer binder.

We think that readers will want to see the microstructures. Particularly, readers with a background in materials science would expect to understand the structure, as it will influence the properties. The images will also help the reader to form a mental picture of the biocoatings and ultimately increase the impact of the research. The section has

been modified to highlight the purpose of the technique (page 9, lines 169-171 and page 10, lines 178-183 and 194-201).

5. The statement in the abstract that PCC 7002 is nonviable in the coatings along with data in Fig 6, showing low ATP levels, is not consistent with the image in Fig 5A, which based on chlorophyll a fluorescence indicates that the cells are viable in coatings. One of the two assays is not accurately reflecting viability.

Thank you for this observation and comment. The CLSM here served to observe the distribution of the bacteria and to provide a simple visual hint at viability. In the text that follows, we indeed show that the viability could not be based on the fluorescence alone. The ATP assay was more reliable to assess the bacterial viability.

In response to this comment, and to provide greater clarity, we have added two sentences to the results and one sentence to the Discussion section (third paragraph):

Page 11, lines 220-222: The CLSM is a qualitative indicator of viability, however it cannot be used as a strict quantitative measure of the viability. Therefore, additional quantitative assays (ATP assays and oxygen evolution measurements) were used.

Page 16, lines 342-344: Therefore, CLSM based on the fluorescence of chlorophyll *a* and phycobilisomes alone should not be used to draw conclusions on the viability of the cyanobacteria.

6. p. 16 line 286: does 10-11 log CFU mean 10¹⁰-11? And if it does, then this seems like a very low number of cells over a m². Moreover, the thickness of the coatings should be mentioned here because a m² isn't a volume.

Yes, the notation 10 – 11 log CFU means 10¹⁰ to 10¹¹. To be more precise, we changed the text to say “in the range from 10 log CFU to 11 log CFU.” (page 12, line 251).

We used a lower number of bacteria to ensure good immobilisation by the binder with reduced bacterial escape. The immobilised *Chroococcidiopsis* 7433 were still able to produce oxygen, even when not at extremely high numbers, and we were able to detect the oxygen production. The thickness of the coatings was already provided in the Methods as 50-70 µm (page 20), but we have now added it after the density statement too (page 12).

The density of cells is expressed per unit area, because that is the figure that is relevant for bioreactors. The number of cells per unit volume is less important.

7. p. 16 lines 289-294. I don't understand this statement. If an experiment was not executed properly, it should be redone - or mention of it removed. Best to remove these sentences.

At equilibrium, the oxygen concentration in the liquid is mathematically related to the oxygen in the vapour phase via Henry's Law. In different replicates, different plateaus could be observed, which is attributed to experimental variability. One possible explanation could be small differences in how the dishes were sealed with Parafilm,

which would allow different concentrations of oxygen to equilibrate in the vapour phase above the liquid.

Note that all experiments presented in Figures 7 and 8 had three replicates.

Upon re-reading the text on lines 289-294 in the original manuscript, we see that it is not clear enough. We have rewritten the description (page 13, lines 257-263).

8. Fig 7 could be removed from the MS. Once you state that strains 7002 (which is dead) and 6803 do not produce oxygen, they should not appear again in the data shown in Fig. 7. The negative control is sufficient. The focus now should be solely on 7433 and all relevant data are presented in Fig. 8

Figure 7 was the first place we showed that the 7002 and 6803 were unable to produce oxygen and we wanted to show that the result is statistically different. We want to include the other species to show how good the extremophile 7433 is by comparison. Panel A in Figure 7 provides some visual evidence for the experimental variability between replicates, whereas Panel B shows the means and SD for the replicates along with a statistical analysis. Note that panel C of Figure 7 shows data for the single species (7433). Panel D shows “escape” from the coatings, which is a different topic. Potentially, we could have seen some cells escaping the coating, even if they were not found to be viable by the ATP assay.

Figure 8 has a different aim. This experiment was run over 30 days, and the data are showing the longevity of the cyanobacteria. The message from Figure 8 builds on the data from Figure 7.

We also think it is important to show the so-called negative data, as these strains are biotechnologically interesting. One of the main objectives of the research is to compare different species in the biocoatings.

9. How does a specific rate of oxygen evolution of .4 g/g biomass per day by *Chroococcidiopsis cubana* embedded in a biocoatings compared to its rate of oxygen evolution in culture?

We performed preliminary oxygen measurements using liquid bacterial cultures, biocoatings and dried bacterial cultures (dried in the same way as the biocoatings), which all contained the same number of bacteria at the beginning of the experiment. The data can be found in the Supplemental Material (Figure S11). The production rate was estimated at the beginning during the linear increase of oxygen production. Indeed, it was possible to observe an initial lag in the oxygen production of the biocoatings compared to the liquid culture. The specific rate of oxygen evolution was around two times lower in biocoatings than in the liquid cultures. This statement was added (page 15, lines 308-313).

10. Discussion. It is important to be more precise about what kind of extremophile *Chroococcidiopsis* is. Extremophile is a broad term.

The *Chroococidiopsis* is a desiccation-resistant extremophile. A statement has now been added in the Abstract and in the first line of the Discussion.

Reviewer #2 (Comments for the Author):

In brief, the study by Krings *et al.* describes encapsulation of cyanobacteria inside biocoatings and comparisons of three cyanobacterial species in regards to survival and photosynthetic activity. Cyanobacteria serve as a significant production platform and therefore it is important to seek for new growth modes, which will be further tailored for specific harvesting of biomass or metabolites. The study provides proof-of-concept for the thermophilic cyanobacterium *Chroococidiopsis cubana* PCC 7433 - it describes particular engineering protocol and demonstrates sustained photosynthetic oxygen evaluation. Overall, the study is nicely done and the manuscript is well written. A few comments and suggestions that need to be addressed are listed below.

We thank the reviewer for their time to read and evaluate our work.

I am concerned about the interpretation of data presented in Fig. 5. The cited paper shows that heated cells lose their red fluorescence. However, not all lethal stresses cause decreased red cyanobacterial fluorescence and vice versa, decreased red cell fluorescence does not necessarily represent cell death. For example, nutrient limited cells actively degrade their Phycobilisomes, the major light harvesting complexes, which emit red fluorescence. Those starved cells are still viable. Indeed, interpretation of red/green fluorescence as an assessment for viability (Fig. 5) is NOT in accordance with the viability determination using CellTiter-Glo (Fig. 6). The three different cyanobacterial species tested exhibit similar red fluorescence but *Synechococcus* is considered non-viable according to the ATP method. In summary of this part - fluorescence data should not be regarded as an indication for viability. The authors may employ the commonly used live/dead SYTOX staining to support the viability findings by ATP measurements. Statements such as in lines 394-396: "While *Synechococcus* 7002 did not survive, *Synechocystis* 6803 and *Chroococidiopsis* 7433 survived the film formation, as determined by viability assays based on CLSM and ATP" need to be revised. Of note, the authors attribute red fluorescence to Chla, which is correct; however, as pointed out above, Phycobilisomes are the predominant pigments under the growth conditions used in this study are the main source of red fluorescence.

We thank the reviewer for these insightful comments and the thorough explanation. The manuscript was adapted to include a mention of phycobilisomes in Section 3.2 (page 11, line 209) and an additional explanation for the experiment was added (page 11, lines 211-213). It was also added that CLSM is a qualitative indicator of viability and cannot be used as a strict measure of the viability, which is why the additional assays were carried out (page 11, lines 219-222).

The text was revised in the third paragraph of the Discussion to indicate that CLSM based on light-harvesting pigments alone was insufficient to draw conclusions on cyanobacterial viability (page 16, Lines 342-344).

Methods indicate that photosynthetic activity was calculated per dry weight (DW) of the bacteria. The cells, however, are embedded within the biocoatings so it is not clear how dry weight of the cells was determined. Please clarify. Is it possible to extract chlorophyll and normalize data to this pigment?

The dry weight of the bacteria added to the biocoatings was estimated at the beginning of the experiment by drying cultures in the stationary phase and weighing them before and after evaporation at 97 °C for 5 h. The mean value of dry weight of 3 – 8 independent cultures was used to calculate the oxygen production per dry weight for all time points. This clarification has been added to the manuscript (page 23, lines 482-486).

If we were to extract the chlorophyll, the solvents would dissolve the coatings and destroy them. It would have been interesting to fabricate coatings for the purpose of chlorophyll extraction alone, however we did not carry out that experiment. Unfortunately, measurements of the absorbance of the biocoatings were not possible because of reduced light transmission.

Lines 291-294: "However, the seal was not fully closed..." Does this mean that may have had higher oxygen evolution rate? Was the seal fully closed in other measurements? Why not repeat? Please clarify.

We calculated the oxygen evolution rate from the rate of increase in dissolved oxygen at the start of the light period. Please see the example in Figure S14 in the Supplemental Material. During this period when the dissolved oxygen level is increasing, oxygen concentrations would also be building in the atmosphere above the liquid. We confirm that all samples were prepared by an identical method. We hypothesise that there was some experimental variability in the light Parafilm seals on the Petri dishes. Hence, the oxygen concentration in the atmosphere above the liquid might have reached different maximum values when comparing the three replicates. That would explain the experimental variability in the plateau value of the dissolved oxygen concentration which – when at equilibrium – is related to the atmospheric concentration via Henry's Law.

Three replicates of all experiments were performed, as are presented in Figures 7 and 8.

We chose this experimental set-up in a Petri dish to allow uniform illumination of all samples from above. but the benefit of the experimental set-up is that is allowed the analysis of the oxygen evolution in real-time over extended periods, without the need to extract a headspace gas periodically.

We now appreciate that the description on lines 291-294 (original manuscript) was clumsy and misleading. The dishes were all lightly sealed with Parafilm, and the phrase "the seal was not fully closed" is not accurate. We have re-written the description (page 13, lines 258-263).

In regards to Figure 4A (f-h): Perhaps it is possible to identify some rod-shaped bacteria in f but I do not see cells in g and h. Use arrow heads to indicate the cells.

Arrows have now been added.

All acronyms should be defined first time they appear in text.

The acronyms have been spelled out when used for the first time.

July 26, 2023

Dr. Suzanne Hingley-Wilson
University of Surrey
Microbial and cellular Sciences
Stag Hill campus
Guildford, Surrey GU3 1DY
United Kingdom

Re: Spectrum01870-23R1 (Oxygen evolution from extremophilic cyanobacteria confined in hard biocoatings)

Dear Dr. Suzanne Hingley-Wilson:

Dear Authors

Reviewer 2 felt that the previous comments were not addressed appropriately. Please address them in full in the next revision

Link Not Available

Sincerely,

Ilana Kolodkin-Gal

Journals Department
Reviewer comments:

Reviewer #1 (Comments for the Author):

This MS is improved. Thank you

Reviewer #2 (Comments for the Author):

I appreciate the changes made to the manuscript and the clarifications. In regards to use of fluorescence as a measure of viability, however, particular parts of the current version still deliver the message that the assay reflects viability. For example, the statement "The CLSM is a qualitative indicator of viability..." (p. 11, lines 225-227; marked version) is misleading.

Fluorescence represents the presence of pigments, regardless of viability. Please rephrase along the line of the discussion on page 16 lines 351-353. Additionally, the relevant Method section was not revised as well as legend to Figure 5, which currently states: "Figure 5. The cyanobacterial strains tested retain viability in the biocoatings." Please go throughout the text and rephrase to avoid delivering a misleading message.

Additional minor point: My request to add arrows to Figure 4 was meant to help identify the cells. Indeed, arrows were added but the Figure legend indicates "Circles, square and arrows indicate points that were analysed by EDX." The revision does not help identifying the cells. Also, why is there a need for different symbols (circles and squares) to indicate regions of analysis?

Staff Comments:

Preparing Revision Guidelines

Please return the manuscript within 60 days; if you cannot complete the modification within this time period, please contact me. If you do not wish to modify the manuscript and prefer to submit it to another journal, please notify me of your decision immediately so that the manuscript may be formally withdrawn from consideration by Microbiology Spectrum.

Response to Reviewers

We thank the reviewer for their time and for their helpful comments. We addressed them individually below. References to page numbers and lines are made in accordance with the marked-up manuscript.

Reviewer #2 (Comments for the Author):

I appreciate the changes made to the manuscript and the clarifications. In regards to use of fluorescence as a measure of viability, however, particular parts of the current version still deliver the message that the assay reflects viability.

For example, the statement "The CLSM is a qualitative indicator of viability..." (p. 11, lines 225-227; marked version) is misleading. Fluorescence represents the presence of pigments, regardless of viability.

Please rephrase along the line of the discussion on page 16 lines 351-353.

Additionally, the relevant Method section was not revised as well as legend to Figure 5, which currently states: "Figure 5. The cyanobacterial strains tested retain viability in the biocoatings."

Please go throughout the text and rephrase to avoid delivering a misleading message.

We thank the reviewer for spotting the unclear message. We changed the mentioned passages (highlighted in yellow):

Abstract on page 2 lines 30-31: Removed the mention of the confocal microscopy and only left ATP assay.

Page 11 line 206: Removed 'and their viability'.

Page 11 lines 208-211: Removed that pigments indicated cell viability or death.

Page 11 lines 220-222: The fluorescence of the pigments can be assessed using CLSM, but cannot be used as a strict quantitative viability measure of viability. Therefore, viability was measured using assays based on ATP and oxygen evolution.

Page 16 lines 338-339: Changed to presence of pigments instead of indication of viability

Page 22 lines 455-457: The mentions of live and dead cells were removed

Page 35 lines 729 and 732: Removed the mention of viability and the cell death

Additional minor point: My request to add arrows to Figure 4 was meant to help identify the cells. Indeed, arrows were added but the Figure legend indicates "Circles, square and arrows indicate points that were analysed by EDX." The revision does not help identifying the cells. Also, why is there a need for different symbols (circles and squares) to indicate regions of analysis?

We agree with the reviewer and thank them for this comment, which makes the figure clearer.

Page 34 lines 720-721: Figure 4 was adapted to include only circles (for latex and halloysite) and arrows (for cyanobacteria), and the legend was modified accordingly.

August 4, 2023

Dr. Suzanne Hingley-Wilson
University of Surrey
Microbial and cellular Sciences
Stag Hill campus
Guildford, Surrey GU3 1DY
United Kingdom

Re: Spectrum01870-23R2 (Oxygen evolution from extremophilic cyanobacteria confined in hard biocoatings)

Dear Dr. Suzanne Hingley-Wilson:

Your manuscript has been accepted, and I am forwarding it to the ASM Journals Department for publication. You will be notified when your proofs are ready to be viewed.

Sincerely,

Ilana Kolodkin-Gal
Editor, Microbiology Spectrum
